# How Is It to Live with Diabetes Mellitus? The Voices of the Diabetes Mellitus Clients

**DOI:** 10.3390/ijerph19159638

**Published:** 2022-08-05

**Authors:** Charity Ngoatle, Tebogo Maria Mothiba

**Affiliations:** 1Department of Nursing Sciences, University of Limpopo, Polokwane 0727, South Africa; 2Faculty of Health Sciences, University of Limpopo, Polokwane 0727, South Africa

**Keywords:** diabetes mellitus clients, diabetes mellitus, voices

## Abstract

***Background*:** Diabetes mellitus is described as a chronic disease resulting from failure of the pancreas to generate enough insulin or inability of the body to efficiently utilize the insulin it generates. Diabetes clients must adjust their lives to live healthy with the diseases for the rest of their lives. Optimizing diabetic knowledge and awareness among people living with diabetes will yield better health outcomes. This study seeks to investigate the knowledge, practices, and challenges of diabetes mellitus clients regarding management of the disease at selected clinics in the Capricorn District of Limpopo Province, South Africa. ***Methods*:** This study used a qualitative research approach and a phenomenological research design. A purposive sampling method was used to acquire the 18 participants for this study. Semi-structured interviews with a guide were used to collect data. Tesch’s coding method was employed for data analysis. ***Results*:** The study findings revealed that there are comparable explanations of what it means to follow medication instructions by diabetes mellitus clients, and challenges living with DM. The findings also indicate that there are problems related to conceptualization of medication instructions among diabetes clients. ***Conclusion*: **This study indicated that diabetes mellitus clients have poor knowledge regarding management of the disease and its process, and problems related to medication instructions. Therefore, proper teaching of clients and guidance regarding diabetes and its management are required to improve compliance and delay of long-term complications.

## 1. Introduction

Diabetes is described as a chronic disease resulting from failure of the pancreas to generate enough insulin (type 1) or (type 1) inability of the body to efficiently utilize the insulin it generates [1]. The International Diabetes Federation (IDF) 2021 report recorded approximately 537 million individuals who are living with diabetes mellitus (DM) in worldwide [2]. The African continent claimed 24 million, with South Africa recording 4.5 million and the numbers continuing to rise [2]. The United Nations’ (UN) Sustainable Development Goal (SDG) number 3.4 aims to reduce premature mortality from communicable diseases by one-third, by preventing, treating, and promoting mental health and well-being [3].

Since diabetes is a chronic disease, the clients must adjust their lives to live healthy with the diseases for the rest of their lives. Optimizing diabetic knowledge and awareness among people will yield better health outcomes for communities [4]. Increased diabetic knowledge is crucial for clients to enhance their lifestyle habits and improve medication adherence, resulting in better health benefits and delayed long-term complications [4]. Diabetes mellitus education is essential not only to the client but also to their families to adjust and manage the required lifestyle modification and offering psychological and dietary support [4].

Offering guidance to diabetes clients on where to source diabetes information is important to obtain credible knowledge. Most diabetes client obtain information from different sources such as health professionals, educational sessions on DM, relatives, journals, television, websites and from other DM patients [5]. It has been found that good knowledge is linked to improved self-management practices and compliance to treatment in clients with diabetes mellitus [6].

Most diabetes clients in Iraq reported poor self-management practices due to a lack of proper knowledge because there are no educational programs on diabetes self-management strategies [7]. In Ethiopia, diabetes clients face challenges such as poor knowledge about diabetes, taking medication correctly, a lack of relationships with health professionals, poor support from friends and resulting loneliness with the disease [8]. Managing diabetes includes taking medication correctly and adopting a healthy lifestyle, which includes exercising and healthy eating [1]. The researchers wanted to understand what is the knowledge, practices and challenges of diabetes mellitus clients regarding management of the disease? This study sort to explore the knowledge, practices and challenges of diabetes mellitus clients regarding management of the disease at selected clinics in the Capricorn District of Limpopo Province, South Africa.

## 2. Methodology

### 2.1. Research Design

A qualitative, phenomenological, explorative, and descriptive research design was used to explore and describe the knowledge, practices, and challenges of clients regarding diabetes treatment at selected Clinics in the Capricorn District of Limpopo Province, South Africa. The design allowed the researcher to obtain an in-depth understanding of diabetic mellitus clients’ knowledge, practices, and challenges of the disease and its treatment.

### 2.2. Study Setting

This study was conducted in four clinics (Dikgale clinic, Seobi-Dikgale clinic, Sebayeng clinic and Makotopong clinic) situated at the Ga-Dikgale village of the Capricorn District in Limpopo Province, South Africa. Dikgale is an established Health and Demographic Surveillance System (HDSS), which is run by the University of Limpopo with a high prevalence of NCDs, hence it was chosen as a study site.

### 2.3. Population and Sampling Strategy

The population of this study comprised all diabetes mellitus clients at Dikgale Village Clinics. The clinics cater for approximately 36,000 people and had 144 diabetes clients on treatment during the study period. Non-probability purposive sampling was used to acquire 18 participants who met the inclusion criteria and agreed to partake in this study. Participants who were free from hearing problems and psychologically sound were included in this study, whereas those who were physically unfit during the data collection period were excluded. The participants included sixteen (16) females and two (2) males. Data were collected in 2019 until saturation was reached.

#### Inclusion and Exclusion Criteria

Clients who were on treatment for more than a month and free from hearing problems were included. This study excluded diabetic clients who were physically unfit during the study period.

### 2.4. Data Collection Procedures

Semi-structured interviews using an interview schedule guide and audiotape were used to collect qualitative data. Field notes were taken to note down non-verbal cues which the voice recorder could not capture. A central question was posed in the same manner to each participant as follows: “**Could you please share with me how you take your medication**”? The central question was followed by probing questions after each participant’s response. Data were collected until saturation, where no new information was coming up from the participants. The participants were also requested to hand in the packets of medications when they were explaining to compare what they were saying with what is written.

### 2.5. Data Analysis

The researchers applied Tesch’s eight-step coding method to analyze the data as suggested by [9]. The researchers listened to the tape repeatedly and transcribed all information verbatim into a script. The transcripts and field notes were read to obtain a sense of the whole. The data were organized into themes and sub-themes. The researchers determined the most descriptive wording for the themes and sub-themes and the researchers re-coded existing material. The themes and sub-themes were summarized, and the data were sent to the independent coder.

### 2.6. Measures to Ensure Trustworthiness

Trustworthiness was ensured by applying the criteria of credibility, dependability, confirmability and transferability [9]. Credibility was ensured through engaging with the participants for approximately thirty minutes and conducting an audit trial. Dependability was ensured by use of the independent coder who is an expert in qualitative research to analyze data and a consensus meeting was held with the researchers to agree on the codes reached independently. Confirmability was ensured by providing a detailed methodology of this study. Transferability was ensured by providing enough details of the research methodology, which entails the research design, the population, the sampling method and the ethical considerations.

### 2.7. Ethical Clearance and Ethical Considerations

The researcher obtained ethical clearance from the University of Limpopo’s Turfloop Research Ethics Committee (TREC), number: TREC/373/2017/PG. The researcher asked permission to conduct this study from the Department of Health Limpopo Province (approval number: LP_2017 11 016), Department of Health Capricorn District and the Nursing manager of the four Dikgale Village Clinics by writing requisition letters attached with research proposal.

The participants agreed to take part in this study voluntarily and were made aware that they could withdraw from participating at any time. Participants were also assured that their identity will not be revealed by assigning them codenames as their identification instead of their names.

## 3. Results

The results are presented below as reflected by diabetes mellitus clients. Table 1 presents the two themes and twelve sub-themes which emerged from this study.

### 3.1. Analogous Explanations of What It Means to Follow Medication Instructions by Diabetes Mellitus Clients

The study participants demonstrated an analogous explanation of what it means to follow medication instructions. This theme is supported by six sub-themes outlined in Table 2 below.

#### 3.1.1. Adherence to Medication Instructions as Directed by Health Professionals

The findings show that participants adhere to medication instructions as directed by health professionals. This is evident in **Participant “D” saying**
*“I take them the way*
*they said*
*should take them, i.e., two times, just that way. I would take them that way; in the morning and the evening. Is it that I would check them*”? **Participant “E” also gave their version,**
**saying** “*It is just the fact that here at the clinic they said I should take the medication 3 times a day then I chose my times that I am going to take them when I wake up, during the day and in the evening before I sleep*”. **Yet Participant “H” said** “*We are guided by the nurses. The nurses teach us day by day that I should not at any time skip the time that I take the medication, and I should not at any time say I forgot them. Meaning lawfully our medication is taken daily and after meals*”.

#### 3.1.2. Questionable Interpretation of Adherence to Medication Instructions 

The study findings revealed that there is a questionable interpretation of adherence to the medications amongst diabetic clients. The interpretation includes the perception of how often the medication should be taken, the frequency explained is not clear and it differs from all the diabetes mellitus clients. This was manifested by **Participant “U” saying** *“I take my pills three times a day. I take them in the morning at 08:00, then at 13:00, and again at 18:00 in the evening”*. **Participant “U” further said**
*“If they say I should take them four times, I should take them at around past 08:00, then at 11:00 I would take them. At 13:00 then I take the medications again and again at 17:00, then I would the medications again. It would be after taking food as it is time for food, then I would be done”*. **Participant “V” gave a diverse opinion but that did not differ much from the previous participant and said** *“My tablets, I take them at 07:00 am, again at 13:00, then lastly at 19:00”*. **Yet Participant “B” said**
*“My diabetic medication, I am taking them in the morning around 08:00 or 09:00 is late and again at night when I go to sleep. I take them two times a day. “We were told the time when we were given the medications here at the clinic to take them in the morning and the evening”*. **Participant “H” also supports the previous participants and said** “*I eat in the morning at 07:00 and take the tablets, at 14:00 I would eat and take my tablets then in the late afternoon around 16:00 to 17:00 I would eat again and take my tablets*”.

#### 3.1.3. Description of the Aspects to Be Considered When following Medication Instructions

The study findings also showed that there are certain aspects to considered when following medication instructions. This was seen in **Participant “C” highlighting that** *“When I take my first pill I eat first; we do not eat too much food because they said we must eat a fist-size pap and then wait for few minutes and take the medication. And they also said we should drink a lot of water so that the pills may be able to melt when they reach the stomach. Is it they say when the pills reach the stomach, they group themselves and sit”?* **Participant “H” also said** *“As I have already said that you give yourself time to say at such and such a time, I would take them even when I visit places, I go with them in my bag. So, when that time arrives, I make sure that I ate something and take my pills so that I would not skip as it is not allowed”*. **Participant “M” gave their version, saying** *“They said these medications should be taken after meals, so after eating we do take them”*. **Lastly, Participant “P” said**
*“I would take them in the morning and then again in the evening. Nonetheless, do not take them on an empty stomach; have something to eat then the pill will follow. That is how the doctor would have told, so you follow what the doctor would have told you”*.

#### 3.1.4. Lack of Adherence to Medication Instructions Is Viewed by DMP as “Digging a Grave for Self”

The findings show that the participants view a lack of adherence to medication as digging a grave for self. This was seen in **Participant “P” saying** *“The complications of not taking medication properly is that when the disease is going to come back to you, is it you would feel you are healed then feel like stopping the pills, most people stop. So, if they stop the pills; the next thing when the disease attacks again it becomes so hard where they would even fall. So, it does not want that when you use it then tomorrow say you no longer want the pill. Just continue until, it is your life, just accept yourself that this pill is your life”*. **Participant “T” said** *“If you do not take them on time, they would not be able to control you well in the body because you would not be taking them on time. You might therefore come and say the pills are not working while the pills are working but the problem being you not taking them correctly. We might suffer dizziness and fall and then have problems. Sometimes your body might itch because you are not using your pills properly. So, it is needful that you use your pills correctly. Take them well in the morning, during the day and in the evening so that you might live well. So, for me to always be complicated it is because of not taking medication correctly, at correct times”*. **Yet Participant “U” alluded that** *“I mean he would be affected badly because if you are not taking medication correctly you are making the disease to grow such that it would not be controlled. Because if you take them the day you like, you are not safe on the medications. It means you are just taking them because you collected them at the clinic while you are supposed to use them correctly and lawfully”*.

#### 3.1.5. Existence of Daily Health Education Sessions in the Clinics Versus Acceptance of Medication Instructions and Related Health Advice as Stipulated by Nurses

The study findings revealed that there are health education sessions taking place daily at clinics. However, the findings also showed that some participants accept and follow medication instructions and related health advice given by nurses, whereas others do not. **Participant “C” said** *“Yes, here at the clinic they are teaching us; after having a prayer, they then start teaching us about medication and say today is the day for Diabetes mellitus, tomorrow is for blood pressure, so on and so forth”*. **Participant “C” also indicated that they accept the health advice given by nurses and said**
*“Is it each time they teach us about sugar diabetes, I listen. Since I am suffering from it, I want to follow the instructions concerning it”*. **Participant “H” further said** *“We are guided by the nurses. The nurses teach us day by day that I should not at any time skip the time that I take the medication, and I should not at any time say I forgot them”*. **Yet Participant “V” also said**
*“We come here at the clinic; here they tell us that the medications they give us, for diabetes mellitus we should take them three times a day. They teach us how we should take them”*. **On the contrary, participants do not follow medication instructions due to different reasons. Participant “D” alluded that** *“We were told that we should take the medication continuously because if we stop, by the time we try them again they might not treat the disease well. But myself now I skipped a month to two because my husband was sick and I and to take care of him”*. **Participant “ZZ” mentioned that** *“I cannot really say I am taking them properly. And yes, the sugar would not be at the required level because I am not taking them properly. Myself I have a machine for testing sugar. So, when I wake up in the morning, I check the sugar and if I find it to be around ke 10.5mmol or 11mmol or around 13mmol there, then, I get discouraged to eat. Because I will eat, and the sugar go a further higher. Then I tell myself not to eat until it goes a little bit down then eat later, because I will never take my medication without eating”*. 

#### 3.1.6. An Explanation That There Is a Need Versus no Need for DMP to Be Assisted with Adherence to the Medications

The findings illustrate that some participants need education on medication instructions, whereas others do not see the need. **Participant “N” indicated that they need assistance and said**
*“Assistance like today we met a certain sister who is assisting us on how to take our medication. So, I feel we need assistance to be reminded of how we should take edications”*. **Participant “O” is aware that they are not taking medication correctly and hence need assistance,** *“Yes, I do need assistance because the way I am taking overdose is not correct”*. **Yet Participant “T” further said**
*“I am not satisfied. I feel I need assistance on how I should eat and how to take medication correctly. I do not have such knowledge, I need it”*. On an obstinate, some participants indicated that they do not need assistance. **Participant “C” said**
*“No, I do not think I need it because our nurses each time we come to collect medication, they give us a health talk about the different diseases and we get educated that people with this kind of disease take their medication like this, those with that disease they take their medication that way, so on and so forth”*. **Participant “V” also said** *“According to me, I do not need it because every day when we collect medications here at the clinic, they teach us how we should take the medication”*. **Yet Participant “U” said**
*“No, I do not need it. I see myself taking the medications correctly, I am satisfied”*. 

### 3.2. Challenges Experienced by Diabetes Mellitus Clients (DMPs)

The participants of this study exhibited that they are experiencing challenges related to medication instructions. This theme is supported by six sub-themes outlined in Table 3 below.

#### 3.2.1. Difficulties in Living with Diabetes Mellitus Co-Existing with Other Body Ailments

The study results revealed that participants have difficulties living with diabetes mellitus while it co-exists with other body ailments. This is evident in **Participant “B”, who indicated that** *“Since I started with this medication, you see these fingers, they just get painful and swollen”*. **Participant “D” also had the same problem and said**
*“Hmn…reality is that we want to be made whole even though when you try getting better other pains rise up”*. **Yet Participant “ZZ” indicated their frustration in this manner and said**
*“**The feet were swelling, and they even changed colour to black for many years until I decided to budget money for specialist. I wish they could stop swelling for good even though they are painful since the other one was once operated, and it is not completely healed, so the other one has to be operated as soon as the other is healed”*.

#### 3.2.2. Socio-Economic Status versus Adherence to Medication

The study results revealed that the socio-economic status of the participants affects adherence to medication. **This is evident in Participant “A”, who said** *“It may be that they say take your medication before you eat but when you have to take the medication you find that there is no food, you going to have to wait for the time there is food, and you eat and then take the medication, then do you see that the medication would not treat you well? Because one day it would reach 10:00 without you taking the medication while you will be waiting for the food then the medication is going to squeeze that one for 13:00”*. **Participant “I” supports this sub-theme and indicated that** *“Yes, we do take our diabetic pills, but my problem is that I would like to know that there were these diabetic pills that we were using on tea, but we are no longer being given them. I just want to know that since we are no longer given the tablets for tea, then could we go back to using sugar? So, “I do not know whether to go back to using sugar or not because those tea tablets were fighting with sexual affairs and now that is a problem”*. **Yet Participant “J” said**
*“Yes, I practice it. Do not you hear me when I say I do not want to lie to say I take them at 08:00 a.m., I could say I take them after each 08:00 a.m. So, I want to tell the truth that sometimes instead of taking the medication at 08:00, I take them after 08:00 a.m. whilst I will be busy with my child or I would be cooking. I would say I want to take the medication at 08:00 am but end up taking it at 09:00 a.m.”*.

#### 3.2.3. Misunderstanding of Medication Instructions and the Effect on Treatment Lifespan

Study findings revealed that participants misunderstood medication instructions and that results in a negative impact on client treatment lifespans. **Participant “M” said** *“The one that is called Metformin, I take it 3 times a day. I take it in the morning at 08:00, during the day at 13:00 and in the evening at 20:00. We just see us drinking them. We are not getting better, when you get heartburn, they say it is the sugar diabetes”*. **Participant “O” avows that** *“The one to be taken three times, I take it at 07:00 am, then at 13:00 and at 19:30. But I do not get better. I have lost a lot of weight; most of the time I lack appetite, I cannot eat but busy taking the pills”*. **Participant “I” also said** “*OH, I also take them after teatime in the morning between 09:00 and 10:00 am., then after lunch around 13:00 to 14:00 and in the evening around 18:00 to 19:00 before I sleep. I am troubled by one thing though, we do take the medications but Hai! They look like they are not working well because the disease just continues. Sometimes you just find yourself walking and you experience cramps or something like that”*.

#### 3.2.4. Lack of Specific Medication Instructions Provided by Professional Nurses

The study results show that there is a lack of specific medication instructions provided by medication dispensers to the participants. **When Participant “G” was asked if it was explained to them how to use the medications, they said** *“No. They have never explained well to me, but they just said I should take the medication in the morning, during the day, and when I go to sleep”*. **Participant “C” also said** *“Myself, I was told to take my medication in the morning after eating and at night before I sleep”*. **Yet Participant “E” said**
*“They said I should take the medication the way they are, but for the times and hours no. They just said in the morning, during the day, and at night”*. **Participant “Y” also added**
*“No, they just say ‘You know that you take you medication twice or thrice’, so if you are taking them twice it means you will take them in the morning and evening and if it is three times, you will take them in the morning, afternoon and evening but the exact time they do not tell us”*. Participant “Y” went further to say that it was never explained to them how they should take the medication, and **Participant “B” said**
*“They just write on the papers to say; once a day, three times a day or two times a day”*.

#### 3.2.5. Lack of Specific Medication Instructions Written on Medication Packages

The study results show that there is a lack of specific medication instructions written on medication packages. **Participant “I” alluded that** *“I usually take them after tea in the morning and in the afternoon after eating, then would take another one, “No, I just see when they have written on the tablets packages; two times a day”*. **Participant “M” also said**
*“The one that is called Metformin, I take it 3 times a day. I take it in the morning, during the day and in the evening. Is it, it is because it is written on the packaging”!*
**Yet Participant “ZZ”, with the same view, said** *“Is it they write on the packages that this one you take it twice a day, the other once daily after meals. They do not tell us the time to say that this one you should take it at 06:00, the other at 20:00. They just say take it twice daily meaning in the morning and evening. If it is three times it means is morning, afternoon, and at night when you sleep, but to say what time, no”*.

#### 3.2.6. Illiterate DMP Not Catered for in Medication Instructions

The results show that illiterate diabetic clients are not catered for, in medication instructions written on the packaging and packet insert. This was specified by **Participant “T”, who indicated that** *“I might not know because I cannot read. I only know the white big one. It is the one that I know I take it three times a day. All the others, they have even given me other pills, but I just do not know if they are related to diabetes mellitus because I am also suffering from high blood pressure. I have two diseases”*. **Another participant, Participant “B”, indicated that they cannot read English but can read the instructions and said**
*“Yes I can, even though I cannot read English. They write the instruction with a pen to say one tablet”*. **Participant “M” also said** *“I do not know. These English things, where would we know them from”?*
**While Participant “T” elaborated** *“Eish! Coming to the times I do not want to lie because I do not know the time, I have*
*just timed that I take it around 09:00 am. Sometimes I forget and take it at 08:00 am*
*or 09:00 am. The next dose I take at 14:00, I time the phone, if I see it written 1 and*
*4 (14), then I start to take it then in the evening when Muvhango starts then I take it*. *Those are the things I use to time my medication times because I cannot read”*.

## 4. Discussion

This study sought to investigate the knowledge, practices, and challenges of clients regarding diabetes treatment. The participants displayed adherence to treatment instructions as directed by medication dispensers. However, the information provided seemed inadequate, and this resulted in most participants not adhering to treatment through a lack of awareness. As a result, those participants did not think they need assistance with interpretation of medication instructions. In health care settings, medication adherence should be taken into account during the prescribing and dispensing process, and it should be given more attention [10]. Therefore, health professionals need to have higher levels of knowledge as they are the primary source of information for patients [4]. However, the authors of [4] found that health professionals in Saudi Arabia had low literacy levels related to diabetes mellitus. On the other hand, low patient knowledge such as stopping medication when feeling better and double dosing were found to be hindrances to treatment adherence [11].

The participants also highlighted some aspects which need to be considered when following treatments which enhance treatment compliance such as taking food before medication. These results concur with [12], where non-adherence to medication was linked to a lack of food as the food was supposed to be taken before medication. Socio-economic status has also been found to play a role in influencing treatment adherence in diabetes clients. Similar results have been recorded, where socio-economic circumstances were amongst the determinants of health in diabetes clients [13].

The participants exhibited a questionable understanding of how they view adherence. The way the participants were following their treatment showed that they do not understand the instructions even though they said they were adhering to them. Similar findings were observed by [14], where clients would leave their clinic visits not understanding health professional instructions related to treatment, even though they said they do. Adherence to treatment includes lifestyle modification, healthy eating and following recommendations related to medication on timing, dosage, frequency, and duration of medication use [15]. In another study, participants would stop medications when feeling better and some were not even aware of the implications thereof [11].

The participants had difficulty living with diabetes while having other body ailments. Similar results have been recorded, where participants had to take many medications for different ailments and ended up not taking diabetes medications as they did not know which one to take when [11].

The results also showed that participants misunderstood medication instructions and the effects on treatment lifespan. The same results were found where participants were following medication instructions but would stop when feeling better, and when they felt that the medication cost (price) was not equivalent to its effectiveness [11].

Some participants understand that non-adherence to treatment instructions is dangerous to their health and life, whereas some do not. Similar results were found where diabetes clients felt that adhering to treatment is difficult and overwhelming even though it was right to do so [16]. Another study found that diabetes clients had a misconception about treatment adherence, which led to non-adherence and poor health outcomes [17].

## 5. Summary

There were challenges faced by the participants in adhering to their treatment. A few aspects are worth mentioning as hindrances to adherence—low socio-economic status, misunderstanding of medication instructions, or non-existence of detailed medication instructions provided by professional nurses or on packages of medication, diabetes mellitus may also coexist with other body ailments, and illiteracy.
The study site is overburdened with communicable diseases which are not controlled despite clients collecting their medications on monthly basis.There was no published data on how patients consume their medication and how professional nurses at primary care explain medication instructions to patients.What is the knowledge, practices, and challenges of diabetes mellitus clients regarding diabetes as a disease, its process and treatment?This study revealed that the medication dispensers at the primary health care level do not give in-depth education regarding diabetes and its treatment to the client.This study recommended that health care professionals should provide in-depth education about diabetes and its treatment, including explaining medication instructions to patients.

## 6. Limitations

This study was conducted in selected clinics in the Capricorn district of the Limpopo province of South Africa. Thus, the findings of this study cannot be generalized to other settings. The same methods, however, can yield similar results.

## 7. Implications of the Findings

The study findings confirm that continuous health education and guidance related to diabetes mellitus and its management should be underway.

### 7.1. The Health Care Professionals

Health professionals dispensing medications to diabetes clients should offer a thorough explanation and clarification of medication instructions to promote adherence. The explanation must include the lifestyle modifications and the disease process.

### 7.2. Limpopo Department of Health

The Limpopo department of health must conduct workshops for health care professionals on the interpretation of medication instructions and evaluate the effectiveness of the available self-management strategies for diabetes.

### 7.3. Department of Education

Health literacy related to interpretation of medication instructions must be incorporated in school curriculums for children to learn how medication is consumed properly and to teach other members at home.

## 8. Conclusions

This study indicated that diabetic clients have deficiencies in knowledge related to diabetes as a disease and treatment such as poor conceptualization of medication instructions and self-management strategies. Proper education regarding the disease and its management, including medication instructions, will improve adherence to treatment.

## Figures and Tables

**Table 1 ijerph-19-09638-t001:** Themes and sub-themes.

Themes	Sub-Themes
Analogous explanations of what it means to follow medication instructions by diabetes mellitus clients	1.1 Adherence to medication instructions as directed by health professionals 1.2 Questionable interpretation of adherence to medication instructions 1.3 Description of the aspects to be considered when following medication instructions 1.4 Lack of adherence to medication instructions viewed as “Digging a grave for self” 1.5 Existence of daily health education sessions in clinics versus acceptance of medication instructions and related health advice as stipulated by nurses 1.6 An explanation that there is a need versus no need for DMP to be assisted with adherence to medications
2. Challenges experienced by DMP	2.1 Difficulties living with diabetes mellitus co-existing with other body ailments 2.2 Socio-economic status versus adherence to medication 2.3 Misunderstanding of medication instructions and the effect on treatment lifespan 2.4 Lack of specific medication instructions provided by professional nurses 2.5 Lack of specific medication instructions written on medication packages 2.6 Illiterate DMP not catered for in medication instructions

**Table 2 ijerph-19-09638-t002:** Analogous explanations of what it means to follow medication instructions by diabetes mellitus.

Theme 1	Sub-Themes
Analogous explanations of what it means to follow medication instructions by diabetes mellitus clients	1.1 Adherence to medication instructions as directed by health professionals 1.2 Questionable interpretation of adherence to medication instructions 1.3 Description of the aspects to be considered when following medication instructions 1.4 Lack of adherence to medication instructions viewed as “Digging a grave for self” 1.5 Existence of daily health education sessions in clinics versus acceptance of medication instructions and related health advice as stipulated by nurses 1.6 An explanation that there is a need versus no need for DMP to be assisted with adherence to medications

**Table 3 ijerph-19-09638-t003:** Challenges experienced by diabetes mellitus clients (DMCs).

Theme 2	Sub-Themes
2. Challenges experienced by DMP	2.1 Difficulties living with diabetes mellitus co-existing with other body ailments 2.2 Socio-economic status versus adherence to medication 2.3 Misunderstanding of medication instructions and the effect on treatment lifespan 2.4 Lack of specific medication instructions provided by professional nurses 2.5 Lack of specific medication instructions written on medication packages 2.6 Illiterate DMP not catered for in medication instructions

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
