# Peer review of "How Is It to Live with Diabetes Mellitus? The Voices of the Diabetes Mellitus Clients"

_ijerph, 2022, doi:10.3390/ijerph19159638_

Round 1

Reviewer 1 Report

The manuscript titled “How is it to live with Diabetes Mellitus? The voices of the diabetes mellitus clients.  This study is innovative, however the authors in the results section could use other statistical analyzes to group their answers.

Discussion is very short, and the authors do not propose the benefits that could be obtained by patients with diabetes, regarding, the practices and challenges of managing the disease

Author Response

  1. Comments - ...authors in the results section could use other statistical analyses to group their answers. Response - the study is qualitative in nature and therefore, followed qualitative data analysis and presentation methods.
  2. Comment - discussion is very short - comment effected as follow: page 10, para 1, line 6, line 11., Page 11, para 1, line 5., Page 11, para 2, line 8., Page 11, para 3., Page 11, para 4., Page 11, last para, line 4.
  3.  

Reviewer 2 Report

Dear Dr. Ngoatle and Dr. Mothiba,

You have conducted an intersting study, one that can be helpful to diabetic patients. However, it needs to have a greater number of participants (an "N" of at least 30) to make its findings statistically relevant. Additionally, I'd explain your inclusionary and exclusionary requirements as well as consider what changes need to made to increase your study's participants beyond the 18 who were selected. It would also benefit your research if you could add more men so your study population has a better gender balance. You might also consider doing this for race, assuming those data are available.

I hope this feedback will help you improve your study for publication.

Author Response

  1. Comment  - an "N" of at least 30, to make the findings statistically relevant. Response - The study is qualitative in nature, therefore, the number of participants was informed by data saturation.
  2. Comment - Explain Inclusion and exclusion criteria. Response -  comment effected under 2.3, line number 4.
  3. Comment - Add more men so your study population has a better gender balance. Response - The population of the study was dominated by women as more men were not coming to the clinics. therefore, only two men consented to the study. At some clinics, there were no men at all.

Thank you so much for the review, it means a lot to the study and future research.

Reviewer 3 Report

Reviewers’ comments to authors manuscript - 1776937
Thank you very much for giving me the opportunity to review this interesting manuscript with the title: How is it to live with Diabetes Mellitus? The voices of the diabetes mellitus clients.

Introduction:

Please check the figure for the population on the African continent. It is given in the manuscript as 24 million. According to my research, it was 240 million in 1950 and currently 1.3 billion people. Is it a typo and a zero is missing? At the same time, please check the actuality of the source, or cite a source here.

Perhaps the sentence is also not quite clear, does the figure mean the indication of people with or without diabetes. Does the number refer to the indication of the number of population or is the indication of people with diabetes on the African continent - also then it would be good to put the number in relation to the number of the total population (prevalence)?

The African continent claimed 24 million, with South Africa recording 4.5 million and the numbers continuing to rise

Please clarify the research question and aim of the study at the end in the introduction. Did you differentiate between type I and type II diabetes in the survey? If not, please indicate.

Method:

During what time period was the survey conducted? Please indicate the year of the survey as well.

What was the age of the 18 interview subjects - was there any other information on the sample - (gender report)?

Please describe the specific procedure of how they determined saturation and how they checked for saturation achievement.

Please briefly describe the role of the researchers involved - as it influences the collection, analysis and interpretation of the data.

The reference to the quality criteria is very helpful, however the procedure for this remains somewhat unclear. Could you please elaborate in more detail on the approach to the four quality criteria under consideration? Thank you very much.

Could you please provide a reference from the literature that shows the analysis method of Renata Tesch. Please show an example of how you proceeded for the analysis - an example from text to category.

Results

Are very insightful and clearly structured and understandable with explication evidenced on the text.

How much time did the interview take?

Please format the text, as in the template.

What health professions are being referred to? Could you please provide the professions with their job titles in addition?

Discussion

Regarding the limitations, please refer to the sample. You had only 2 male participants. What was the reason for that?

This is an important paper and very understandably written and further results for the target group and the spread of diabetes. The manuscript can with the few comments, increase the significance a bit more. I would like to motivate the authors to edit the manuscript for the recommendations

Author Response

Thank you for the comments.

  1. introduction - comment on population. Response = the statistics are correct according to the source provided in the introduction. comment = clarify the research question and the aim of the study at the end in the introduction. Response = comment effected, see page 2, last para under introduction. comment = did you differentiate between type I and type II diabetes in the survey? response= no. the study did not have interest in any type of diabetes but diabetes in general.
  2. Method. comment = during what period was the survey conducted? please indicate the year of the survey as well.  Response = year of study effected. check page 2, under population and sampling, line 7. comment = what was the age of the 18 interview subjects? response = the age was not asked. and there was no additional information on the sample which was of interest to the researchers/ study. Comment= please describe the specific procedures of how they determine saturation.  response= page 3, para 1, line 5. comment= please describe the role of the researchers involved. response= 1st researcher was the one collecting data and the main roleplayer in the study, 2nd researcher is the supervisor and was overseas during the whole study process.  comment =The reference to the quality criteria is very helpful, however the procedure for this remains somewhat unclear. Could you please elaborate in more detail on the approach to the four quality criteria under consideration? response = Credibility

    The following steps were taken to ensure credibility: 1. Providing a detailed description of the methods used to collect and analyze the data. Secondly, long-term engagement with participants, where the researcher has spent four months collecting data on the ground. Thirdly, the researcher triangulated the data using different data collection methods, including document analysis and semi-structured interviews. Fourthly, the researcher presented the data at conferences and workshops for peer review and academic scrutiny. Finally, participants were informed that they could withdraw from the study, if they so chose, so that they can willingly participate and provide honest information.

    Dependability

    To ensure dependability, raw data was compiled, the data collection process followed, the analysis products, the process notes, and the reflection of the researcher and the scrutiny of the supervisors of the study. Also, the sampling method determines the reliability of the data. An extensive explanation of the study's sampling method was provided by the researcher. To enable future researchers to duplicate the work even if not necessarily to achieve the same results, the researcher provided detailed descriptions of the data collection and analysis methods.

    Confirmability

    Raw data were provided in this study to ensure confirmability. In asserting that the information was drawn from the participants, the researcher did not present any data that wasn't provided by them. Furthermore, an independent coder was involved and all data collection products were made available as evidence. Moreover, the researcher provided a detailed description of the methodology used, in order to enable the reader to assess the validity of the data. The supervisor also conducted the audit trail.

    Transferability

    As a result of the sampling and data collection methods used in this study, the extent to which the findings may be applicable to other individuals and situations can be determined. This study involved Diabetes mellitus patients on treatment in four clinics. However, only the researcher took part in data collection, which involved 18 participants.

     comment = Could you please provide a reference from the literature that shows the analysis method of Renata Tesch. Please show an example of how you proceeded for the analysis - an example from text to category. response =

    Steps     Procedure

    1. Firstly, the researcher  listened  to  the  recorded  interviews  and  transcribed  the information verbatim. The entire transcripts were then read carefully to obtain a sense of the whole and some ideas were written down.

    2.  One interview was selected and read to get the information, writing down thoughts that  came to mind. A table was made with all the topics and sub-topics that emerged but there were not grouped. The researcher took another transcript, read it trying to relate it with the first one. Other topics sub-topics emerged and were added to the previous ones.

    3. The researcher then made a list of  all the topics. Similar topics were grouped to form  themes and sub-themes. The themes and the sub-themes were then named using words that best described all the grouped. Where necessary, the themes were changed into sub-themes and the sub-themes also were rearranged as themes.

    4. The themes were abbreviated as codes, which were written next to the  appropriate segments of the transcripts. The researcher prepared this preliminary by organising schemes to see whether new themes and codes emerged. Whenever a new sub-theme emerged, it was added to the appropriate theme.

    5.         The researcher came up with the most descriptive wording for the themes and sub-themes. For example, misinterpretation, non-compliance, and double dosage. The

    comment = how much time did the interview take? Response = 18 to 49 minutes.  Comment =

    comment = Please format the text, as in the template. response = effected

    comment = What health professions are being referred to? Could you please provide the professions with their job titles in addition? Registered Primary health care nurses.

    Discussion

    comment = Regarding the limitations, please refer to the sample. You had only 2 male participants. What was the reason for that? response = addressed in the 2 previous reviewers.

Round 2

Reviewer 2 Report

Dear Sirs/Mesdames, This is an improvement over your last version but it still needs some English grammar adjustments, e.g., Abstract, the study sort should be seeks. For the population you studied (18), your results seem consistent but I'd like to see a larger "n" studied (at least 30). Best, DrS

Author Response

Comment - English grammar adjustments, e.g., Abstract, the study sort should be seeks.

Response - comment effected in the abstract, line 5. Other English grammar adjustments were also done in the abstract. Please refer to the abstract tract changes.

Comment - For the population you studied (18), your results seem consistent but I'd like to see a larger "n" studied (at least 30).

Response - The researchers conducted qualitative research on this study. And with qualitative, the number of participants is informed by data saturation. We reached data saturation at participant 18 and the data collection took place around 2019/ 2020. I do not understand why we need 30 participants as this is not a quantitative study and we do not seek to quantify anything.
